# Calibration Method for Angular Positioning Deviation of a High-Precision Rotary Table Based on the Laser Tracer Multi-Station Measurement System

**Hongfang Chen [1],\*, Bo Jiang [1], Hu Lin [2], Shuang Zhang [1], Zhaoyao Shi [1], Huixu Song [1] and Yanqiang Sun [1]**

[1] Beijing Engineering Research Center of Precision Measurement Technology and Instruments (Beijing University of Technology), Beijing 100124, China
[2] National Institute of Metrology, Beijing 102200, China
\* Correspondence: chf0302@126.com; Tel.: +86-13401078692

**Abstract:** This paper proposes a calibration method for angular positioning deviation of a high-precision rotary table based on the laser tracer multi-station measurement system. The algorithm error of the calibration method for angular positioning deviation of a high-precision rotary table based on the laser tracer multi-station measurement system was mainly discussed. During the experiments, the laser tracer was fixed on the work surface of the rotary table, and the rotary was fixed on the work surface of the coordinate measurement machine (CMM). The rotary table was rotated with the same angular interval. In this case, an optimization method for calculating the coordinates of a laser tracer station by using Levenberg–Marquardt algorithm and singular value decomposition transform was proposed. Then, the angular positioning deviation of the rotary table was calibrated by an established geometric relationship model between the coordinates of laser tracer stations and the rotation angle of the rotary table. The angular positioning deviation of the high-precision rotary table was as low as ±0.9″, and the error of the calibration method was ±0.4″. The experimental results proved the feasibility of the proposed calibration method. The calibration method proposed in this paper is suitable for the case that the rotary table is not linked with the CMM, especially for large high-precision rotary tables.

**Keywords:** angular positioning deviation; calibration method; laser tracer multistation; measurement system; rotary table

## 1. Introduction

In the modern manufacturing industry, due to the processing complexity of parts, many parts are processed by the angular positioning deviation of the rotary table in order to meet the position accuracy requirements at different angles. The angular positioning deviation of the rotary table directly affects the machining accuracy of parts [1,2]. In specific machining applications, the rotary table is not linked to the machine tool. On the contrary, the rotary table is horizontally mounted on the working platform of the machine tool. Through the joint drive with the three coordinates of X, Y, and Z axes, the parts can be rotated within 360° with the same angular interval to achieve the processing of complex parts.

In order to ensure the accuracy of the angular positioning deviation of the rotary table, it is necessary to calibrate the angular positioning deviation of the rotary table. Many calibration methods of rotary tables were reported. Ming Zeng et al. [3] proposed a high-precision turret angle measurement system calibration method, which could avoid the measurement and calculation errors caused by human factors through the photoelectric autocollimator and multiface prism method. Tao Xu et al. [4]

proposed a calibration method of target tracking rotary table based on coordinate transformation. By establishing a precise calibration model of the rotary table, the experimental results showed that the tracking accuracy of the vehicle-borne opto-electronic tracking system can be improved to 0.1°. Mian Wang et al. [5] studied the theodolite calibration method, experimentally analyzed the error factors of the calibration system, and improved the calibration accuracy of the angular displacement of large rotary tables. Pujun Bai et al. [6] proposed a simple and efficient calibration method for the rotation angle of a single-degree-of-freedom rotary table based on a laser tracker, and established the relationship between the spatial position and the angle. Its angular positioning deviation is reduced to 5″. Shiwei Pi et al. [7] proposed a geometric error detection method for rotary feed drive, which mainly used a laser interferometer and plane mirror instead of photoelectric autocollimator and multiface prism to improve the geometric accuracy. The experimental results showed that the angular positioning deviation can reach 4.77″. Yongyao Yan et al. [8] proposed a new method of adjusting error compensation based on angle calibration, which compensated the adjustment error of photoelectric autocollimator, and angular positioning deviation is as low as 0.922″. The experiment proved the rapidity and reliability of the method.

However, the above-mentioned calibration method for angular positioning deviation of the rotary table involves high-precision optical instruments such as a laser interferometer and a photoelectric autocollimator and involves the construction of an optical path, which takes a long time and cannot meet the requirements of fast and efficient calibration. The method of calibrating the rotary table by mounting optical components such as multisurface prism on the spindle is not suitable for large rotary tables, because it is difficult to install a multisurface prism with the high installation accuracy. In the method of calibrating the rotary table by the laser tracker, because the tracking angle range of the laser tracker is small, the full circumference measurement cannot be realized. The electronic theodolite calibration method can avoid the difficulties in determining the rotation center of an eccentric rotary table and the small field of view of the autocollimator, but it is difficult to select the observation points and the calibration accuracy is not high.

This paper proposes a calibration method for angular positioning deviation of a high-precision rotary table based on the laser tracer multi-station measurement system. The mapping relationship between the coordinate information of laser tracer and rotation angle of the rotary table was established based on the high-precision relative interference length of the laser tracer, so as to realize the calibration of the angular positioning deviation of the rotary table. The algorithm error of calibrating the high-precision rotary table based on the laser tracer multi-station measurement system were analyzed. The calibration method was experimentally verified.

## 2. Laser Tracer Multi-Station Measurement System

The laser tracer multi-station measurement model is shown in Figure 1. The rotary table is mounted on the working plane of the CMM (coordinate measurement machine), and the laser tracer is placed on the rotary table. The multi-station measuring system of the laser tracer consists of CMM, rotary table, and laser tracer. The coordinate system of the rotary table is the same as the coordinate system of the CMM, as shown in Figure 2. The cat eye reflector of the laser tracer [9–11] is fixed on the probe of the CMM as the measured point. The cat eye reflector has the same trajectory as the probe of the CMM. When the CMM controls the probe to move in the measurement space, the cat eye reflector also follows the probe of the CMM. The laser beam from the laser tracer is incident on the cat eye reflector and reflected back to the tracking head of the laser tracer. After receiving the reflected beam from the cat eye reflector, the laser tracer measures the relative displacement between the cat eye reflector (i.e., the measured point) and the laser tracer. The CMM provides the coordinates of measured points.

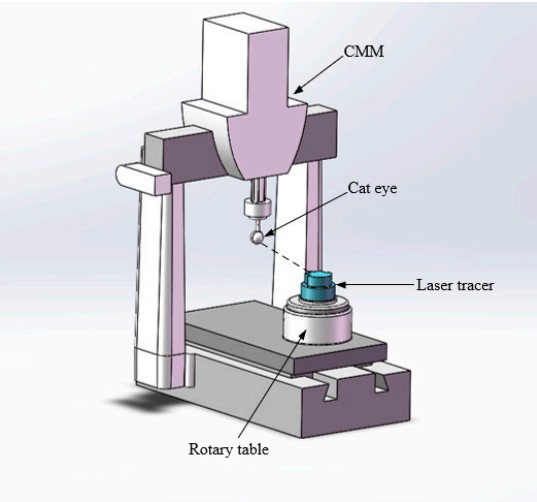

**Figure 1.** Laser tracer multi-station measurement model. CMM: coordinate measurement machine.

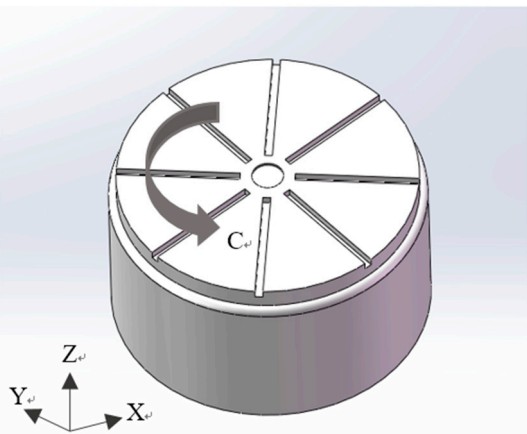

**Figure 2.** Schematic diagram of the rotary table coordinate system.

It is assumed that the laser tracer is used to measure $n$ measured points at $m$ stations. The coordinates of laser tracer stations are $P_j(X_j,Y_j,Z_j)$; the coordinates of measured points are $A_i(x_i,y_i,z_i)$; the distance from the laser tracer at each station $P_j$ to the initial measured point $A_1$ is $d_j$; the high-precision interferometric length measured by the laser tracer is $l_{ij}$. According to the distance formula of two points in three-dimensional (3D) space, the following relationship is established. The distance $d_{ji}$ of the laser tracer from each station to the measured point is obtained,

$$d_{ji} = \sqrt{\left(x_i - X_j\right)^2 + \left(y_i - Y_j\right)^2 + \left(z_i - Z_j\right)^2} = d_j + l_{ij} \tag{1}$$

where $i = 1,2,3 \ldots n, j = 1,2,3, \ldots m$.

Since $n$ measured points are measured at $m$ stations, according to Equation (1), the number of unknowns is $4m + 3n$. In order to solve Equation (1), the following relationship should hold:

$$m \times n \geq 4m + 3n \tag{2}$$

From Equation (2), we get $m \geq 4, n \geq 16$.

With the high-precision interferometric length measured by the laser tracer as the constraint condition, Equation (1) can be rewritten as the error equation,

$$v_{ij} = \sqrt{\left(x_i - X_j\right)^2 + \left(y_i - Y_j\right)^2 + \left(z_i - Z_j\right)^2} - d_j - l_{ij} \tag{3}$$

Since Equation (3) is a nonlinear equation, it is difficult to directly solve it. According to a linearization method, the nonlinear function is converted into a linear function, which is then processed according to linear parameters. Let $\sqrt{\left(x_i - X_j\right)^2 + \left(y_i - Y_j\right)^2 + \left(z_i - Z_j\right)^2}$ be recorded as $L_{ij}$, and the first-order Taylor series expansion of Equation (3) is obtained as,

$$v_{ij} = L_{ij}|_0 + \frac{x_i|_0 - X_j|_0}{L_{ij}|_0} \cdot \left(\Delta x_i - \Delta X_j\right) + \frac{y_i|_0 - Y_j|_0}{L_{ij}|_0} \cdot \left(\Delta y_i - \Delta Y_j\right) + \frac{z_i|_0 - Z_j|_0}{L_{ij}|_0} \cdot \left(\Delta z_i - \Delta Z_j\right) - d_j - l_{ij} \tag{4}$$

where $L_{ij}|_0 = \sqrt{\left(x_i|_0 - X_j|_0\right)^2 + \left(y_i|_0 - Y_j|_0\right)^2 + \left(z_i|_0 - Z_j|_0\right)^2}$; the mark symbol $|_0$ is the approximate of this value, and the approximate values $x_i|_0$, $y_i|_0$ and $z_i|_0$ of the measured points in the actual calculation are replaced by the measured points in the coordinate system of CMM. Based on the singular value decomposition algorithm and the precision control threshold [12], according to Equation (4), the correction values $(\Delta x_i, \Delta y_i, \Delta z_i)$ of the measured points in CMM can be obtained.

## 3. Calibration Method for the Angular Positioning Deviation of the Rotary Table

### 3.1. Self-Calibration of Laser Tracer Station Based on Levenberg–Marquardt Algorithm

3.1.1. Self-Calibration Method

According to Equation (1), let

$$f_i\left(X_j, Y_j, Z_j, d_j\right) = \sqrt{\left(x_i - X_j\right)^2 + \left(y_i - Y_j\right)^2 + \left(z_i - Z_j\right)^2} - d_j - l_{ij} \tag{5}$$

Record $f_i(x) = (f_1(x), f_2(x), \ldots, f_n(x))$, then

$$\min_{x \in \mathbf{R}^n} F(x) = \frac{1}{2} \sum_{i=1}^{n} f_i^2(x) = \frac{1}{2} \|f(x)\|^2 \tag{6}$$

where $\mathbf{R}^n$ is a set of $n$-dimensional real numbers and $n$ is the number of measured points. The gradient of the objective function $F$ is denoted as $g(x)$, and we can obtain

$$g(x) = \nabla F(x) = \nabla\left(\frac{1}{2}\|f(x)\|^2\right) = J(x)^{\mathrm{T}} f(x) = \sum_{i=1}^{n} f_i(x) \nabla f_i(x) \tag{7}$$

where $J(x) = f'(x) = \left(\nabla f_1(x), \nabla f_2(x), \ldots \nabla f_n(x)\right)^{\mathrm{T}}$.

In this paper, Levenberg–Marquardt algorithm (L-M) is used for iteration, and the coordinates of laser tracer stations and the distance from laser tracer stations to the initial measured point are obtained by iteration. The search direction of the iteration is set as $h_i$, then

$$h_i = \underset{d \in \mathbf{R}^n}{\mathrm{argmin}} \|J_i h + f_i\|^2 + \mu_i \|h\|^2 \tag{8}$$

where $\mu_i > 0$; $\mu_i$ is the positive parameter introduced to adjust the search direction; $f_i$ is the set of error equations; $J_i$ is the gradient matrix of error equations; $h$ is the set of search directions.

By optimality condition [13,14], $h_i$ satisfies

$$\nabla\left(\|J_i h + f_i\|^2 + \mu_i \|h\|^2\right) = 2\left[\left(J_i^{\mathrm{T}} J_i + \mu_i I\right) h + J_i^{\mathrm{T}} f_i\right] = 0 \tag{9}$$

where $I$ is $n \times n$ identity matrix.

After solving Equation (9), we can get

$$h_i = -\left(J_i^{\mathrm{T}} J_i + \mu_i I\right)^{-1} J_i^{\mathrm{T}} f_i \tag{10}$$

where

$$J_i = J_i\left(X_j, Y_j, Z_j, d_j\right) = \begin{bmatrix} \frac{\partial f_1}{\partial X_j} & \frac{\partial f_1}{\partial Y_j} & \frac{\partial f_1}{\partial Z_j} & \frac{\partial f_1}{\partial d_j} \\ \frac{\partial f_2}{\partial X_j} & \frac{\partial f_2}{\partial Y_j} & \frac{\partial f_2}{\partial Z_j} & \frac{\partial f_2}{\partial d_j} \\ \vdots & \vdots & \vdots & \vdots \\ \frac{\partial f_n}{\partial X_j} & \frac{\partial f_n}{\partial X_j} & \frac{\partial f_n}{\partial X_j} & \frac{\partial f_n}{\partial X_j} \end{bmatrix}_{n\times 4}$$

$$= \begin{bmatrix} \frac{X_j-x_1}{\sqrt{\left(x_1-X_j\right)^2+\left(y_1-Y_j\right)^2+\left(z_1-Z_j\right)^2}} & \frac{Y_j-y_1}{\sqrt{\left(x_1-X_j\right)^2+\left(y_1-Y_j\right)^2+\left(z_1-Z_j\right)^2}} & \frac{Z_j-z_1}{\sqrt{\left(x_1-X_j\right)^2+\left(y_1-Y_j\right)^2+\left(z_1-Z_j\right)^2}} & -1 \\ \frac{X_j-x_2}{\sqrt{\left(x_2-X_j\right)^2+\left(y_2-Y_j\right)^2+\left(z_2-Z_j\right)^2}} & \frac{Y_j-y_2}{\sqrt{\left(x_2-X_j\right)^2+\left(y_2-Y_j\right)^2+\left(z_2-Z_j\right)^2}} & \frac{Z_j-z_2}{\sqrt{\left(x_2-X_j\right)^2+\left(y_2-Y_j\right)^2+\left(z_2-Z_j\right)^2}} & -1 \\ \vdots & \vdots & \vdots & \vdots \\ \frac{X_j-x_n}{\sqrt{\left(x_n-X_j\right)^2+\left(y_n-Y_j\right)^2+\left(z_n-Z_j\right)^2}} & \frac{Y_j-y_n}{\sqrt{\left(x_n-X_j\right)^2+\left(y_n-Y_j\right)^2+\left(z_n-Z_j\right)^2}} & \frac{Z_j-z_n}{\sqrt{\left(x_n-X_j\right)^2+\left(y_n-Y_j\right)^2+\left(z_n-Z_j\right)^2}} & -1 \end{bmatrix}_{n\times 4}$$

Let $m_i'$ be the smallest non-negative integer satisfying Equations (11)–(14), then

$$f\left(X_j + \beta^{m'} h_i\right) \le f\left(X_j\right) + \sigma\beta^{m'} g_i^{\mathrm{T}} h_i \tag{11}$$

$$f\left(Y_j + \beta^{m'} h_i\right) \le f\left(Y_j\right) + \sigma\beta^{m'} g_i^{\mathrm{T}} h_i \tag{12}$$

$$f\left(Z_j + \beta^{m'} h_i\right) \le f\left(Z_j\right) + \sigma\beta^{m'} g_i^{\mathrm{T}} h_i \tag{13}$$

$$f\left(d_j + \beta^{m'} h_i\right) \le f\left(d_j\right) + \sigma\beta^{m'} g_i^{\mathrm{T}} h_i \tag{14}$$

where $\sigma \in (0,1)$ and $\beta \in (0,1)$.

In order to ensure that $h_i$ is the descending direction of $f_i(x)$ at $x_i$, the initial value of $\mu_i$ is set at the beginning of iteration, and $\mu_i$ is adjusted continuously by calculating $h_i$. According to the allowable error $\varepsilon$ of the laser tracer multi-station measurement system, the coordinates $P_j(X_j,Y_j,Z_j)$ of laser tracer stations and the distance $d_j$ from laser tracer stations to the initial measured point can be calibrated by iteration.

3.1.2. Selection of Parameter $\mu_i$

The key of L-M algorithm is to select the parameter $\mu_i$. According to the current iteration point, the quadratic function is assumed to be

$$q_i(h) = F_i + \left(J_i^{\mathrm{T}} f_i\right)^{\mathrm{T}} h + \frac{1}{2} h^{\mathrm{T}} \left(J_i^{\mathrm{T}} J_i\right) h \tag{15}$$

where $F_i$ is the objective function.

The incremental ratio of objective function to quadratic function is expressed as $r_i$, then we can get

$$r_i = \frac{\Delta F_i}{\Delta q_i} = \frac{F(x_i + h_i) - F(x_i)}{q_i(h_i) - q_i(0)} = \frac{F(x_{i+1}) - F(x_i)}{\left(J_i^{\mathrm{T}} f_i\right)^{\mathrm{T}} h_i + 0.5 h_i^{\mathrm{T}} \left(J_i^{\mathrm{T}} J_i\right) h_i} \tag{16}$$

when $r_i$ approaches 0 or 1, this parameter needs to be adjusted [15]. Generally, the critical values of $r_i$ are 0.25 and 0.75, so the selection rule of parameter $\mu_i$ is obtained as

$$\mu_{i+1} = \begin{cases} 0.1\mu_i, & r_i > 0.75 \\ \mu_i, & 0.25 \leq r_i \leq 0.75 \\ 10\mu_i, & r_i < 0.25 \end{cases} \tag{17}$$

In the iteration process, the initial value of $\mu_i$ is given, and the value of each iteration step is taken as the initial value of the next iteration. According to the calculated $h_i$ and $r_i$, the parameter $\mu_i$ is selected. Then, according to the selected parameter $\mu_i$, $h_i$ is calculated, and the linear search is carried out to complete the iteration process.

### 3.2. Optimizing Coordinates of Laser Tracer Stations

In order to improve the accuracy of the coordinates of laser tracer stations obtained by the self-calibration algorithm, the Singular Value Decomposition (SVD) transformation [16,17] of the covariance matrix is used for plane fitting. The coordinates of $m$ laser tracer stations obtained by self-calibration algorithm are fitted into a plane. In the fitting plane, the distances from the coordinates of $m$ laser tracer stations to the fitting plane should have the minimum residuals. The optimized coordinates of laser tracer stations can be obtained by projecting $m$ coordinates of the laser tracer onto the fitting plane.

The principle of SVD of covariance matrix is provided as

$$A = U\Sigma V^{\mathrm{T}} \tag{18}$$

where $U$ is the left singular orthogonal vector matrix; $\Sigma$ is the diagonal singular value matrix; $V$ is the right singular orthogonal vector matrix.

The fitting plane is set as

$$aX + bY + cZ + e = 0 \tag{19}$$

The singular vector corresponding to the minimum singular value is the normal vector direction $\vec{n} = (a, b, c)$ of the fitting plane equation.

By solving SVD, the fitting plane equation coefficients $a$, $b$, $c$, and $e$ can be obtained, and then the fitting plane equation $aX + bY + cZ + e = 0$ is obtained.

The optimized coordinates of laser tracer stations are set as $P_j{}'\left(X_j{}', Y_j{}', Z_j{}'\right)$. According to the parallelism between straight line $P_j P_j{}'$ and normal vector $\vec{n} = (a, b, c)$ of the plane $aX + bY + cZ + e = 0$, the parametric equation of straight line $P_j P_j{}'$ is

$$\begin{cases} X'_j = X_j + a\lambda \\ Y'_j = Y_j + b\lambda \\ Z'_j = Z_j + c\lambda \end{cases} \tag{20}$$

After substituting $\left(X_j{}', Y_j{}', Z_j{}'\right)$ into plane equation $aX + bY + cZ + e = 0$, we can get

$$\lambda = -\frac{aX'_j + bY'_j + cZ'_j + e}{a^2 + b^2 + c^2} \tag{21}$$

After substituting $\lambda$ into Equation (20), the optimized coordinates $P_j{}'\left(X_j{}', Y_j{}', Z_j{}'\right)$ of laser tracer stations are obtained.

*3.3. Calibration of The Angular Positioning Deviation of the Rotary Table*

3.3.1. Solving the Center of the Circle of the Laser Tracer Station Coordinate Fitting Circle

Let the plane fitted by the *m* station coordinates of the laser tracer obtained by the self-calibration algorithm be $\gamma$. The center of the circle fitted by m station coordinates of laser tracer is $O(x_c,y_c,z_c)$. The projection equation of the plane formed by the plane $\gamma$ at plane $z = z_c$ is set.

$$\left(X_j' - x_c\right)^2 + \left(Y_j' - y_c\right)^2 = R^2 \tag{22}$$

where $x_c$ is the coordinate value of the center of the fitting circle in the X-axis direction; $y_c$ is the coordinate value of the center of the fitting circle in the Y-axis direction, and $R$ is the radius of the fitting circle.

According to the principle of nonlinear least squares, the objective function can be defined as

$$f = \sum_{j=1}^{m}\left((X_j' - x_c)^2 + (Y_j' - y_c)^2 - R^2\right)^2 \tag{23}$$

Let $g\left(X_j', Y_j'\right) = (X_j' - x_c)^2 + (Y_j' - y_c)^2 - R^2$, then

$$f = \sum_{j=1}^{m} g\left(X_j', Y_j'\right)^2 \tag{24}$$

In order to minimize the objective function *f*, Equation (24) should satisfy the following conditions,

$$\begin{cases} \frac{\partial f}{\partial R} = -4R \sum_{j=1}^{m} g\left(X_j', Y_j'\right) = 0 \\ \frac{\partial f}{\partial x_c} = -4 \sum_{j=1}^{m} g\left(X_j', Y_j'\right)\left(X_j' - x_c\right) = -4 \sum_{j=1}^{m} X_j' g\left(X_j', Y_j'\right) = 0 \\ \frac{\partial f}{\partial y_c} = -4 \sum_{j=1}^{m} g\left(X_j', Y_j'\right)\left(Y_j' - y_c\right) = -4 \sum_{j=1}^{m} Y_j' g\left(X_j', Y_j'\right) = 0 \end{cases} \tag{25}$$

Let $u_j = X_j' - \overline{X}'$, $u_c = x_c - \overline{X}'$, $v_j = Y_j' - \overline{Y}'$ and $v_c = y_c - \overline{Y}'$, where $\overline{X}' = \frac{1}{m} \sum_{j=1}^{m} X_j'$, $\overline{Y}' = \frac{1}{m} \sum_{j=1}^{m} Y_j'$, then we can get

$$\begin{cases} \sum_{j=1}^{m} u_j g\left(u_j, v_j\right) = 0 \\ \sum_{j=1}^{m} v_j g\left(u_j, v_j\right) = 0 \end{cases} \tag{26}$$

Based on Equation (26), we can get

$$\begin{cases} \sum_{j=1}^{m} \left(u_j^3 - 2u_j^2 u_c + u_j v_j^2 - 2u_j v_j v_c\right) = 0 \\ \sum_{j=1}^{m} \left(u_j^2 v_j - 2u_j v_j u_c + v_j^3 - 2v_j^2 v_c\right) = 0 \end{cases} \tag{27}$$

According to Equation (27), we can get

$$
\begin{cases}
u_c = \dfrac{\sum\limits_{j=1}^{m} u_j^2 v_j \sum\limits_{j=1}^{m} u_j v_j - \sum\limits_{j=1}^{m} u_j^3 \sum\limits_{j=1}^{m} v_j^2 - \sum\limits_{j=1}^{m} u_j v_j^2 \sum\limits_{j=1}^{m} v_j^2 + \sum\limits_{j=1}^{m} u_j v_j \sum\limits_{j=1}^{m} v_j^3}{2\left(\left(\sum\limits_{j=1}^{m} u_j v_j\right)^2 - \sum\limits_{j=1}^{m} u_j^2 \sum\limits_{j=1}^{m} v_j^2\right)} \\[4ex]
v_c = \dfrac{-\sum\limits_{j=1}^{m} u_j^2 \sum\limits_{j=1}^{m} u_j^2 v_j + \sum\limits_{j=1}^{m} u_j^3 \sum\limits_{j=1}^{m} u_j v_j + \sum\limits_{j=1}^{m} u_j v_j \sum\limits_{j=1}^{m} u_j v_j^2 - \sum\limits_{j=1}^{m} u_j^2 \sum\limits_{j=1}^{m} v_j^3}{2\left(\left(\sum\limits_{j=1}^{m} u_j v_j\right)^2 - \sum\limits_{j=1}^{m} u_j^2 \sum\limits_{j=1}^{m} v_j^2\right)}
\end{cases}
\tag{28}
$$

Then

$$
x_c = u_c - \overline{X}'
\tag{29}
$$

$$
y_c = v_c - \overline{Y}'
\tag{30}
$$

According to Equation (19), we can get

$$
z_c = -\frac{ax_c + by_c + d}{c}
\tag{31}
$$

In summary, the coordinates of the center of the circle fitted by the laser tracer $O(x_c, y_c, z_c)$ are obtained.

### 3.3.2. Calibration of the Angular Positioning Deviation of the Rotary Table

As shown in Figure 3, the angle $\theta_j'$ between adjacent stations of the laser tracer is

$$
\theta'_j = \arccos\frac{|P_{j-1}'O|^2 + |P_j'O|^2 - |P_{j-1}'P_j'|^2}{2|P_{j-1}'O| \cdot |P_j'O|}
\tag{32}
$$

where $P_{j-1}'$ is the coordinate of the j-th station optimized by the laser tracer; $P_j'$ is the coordinate of the j-th station optimized by the laser tracer, and $\theta_j'$ is the actual rotation angle of the rotary table.

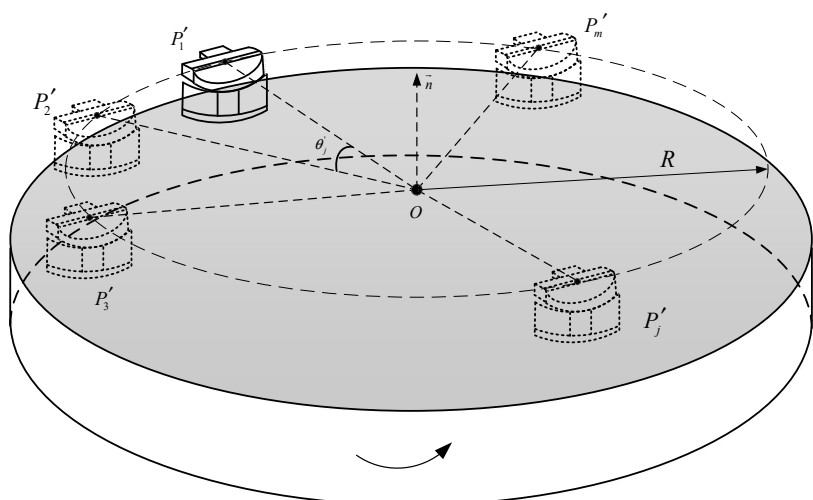

**Figure 3.** Schematic diagram of the relationship between the position of the laser tracer and the rotating axis.

Set the theoretical rotation angle of the rotary table as $\theta_j$. When the number of the coordinates of the laser tracer stations is $m$, the rotary table rotates $(m - 1)$ times. According to Equation (32), the angular positioning deviation $\Delta\theta_j$ is obtained

$$\Delta\theta_j = \theta'_j - \theta_j \tag{33}$$

where $\theta_j$. is the angle of theoretical rotation of the rotary table, $j = 1,2,3 \dots m - 1$.

According to the multimeasurement averaging method, the angular positioning deviation $\overline{\sigma}_{\Delta\theta}$ of the rotary table based on the laser tracer multi-station measurement system is calibrated as

$$\overline{\sigma}_{\Delta\theta} = \pm\frac{1}{2}\sqrt{\frac{\sum\limits_{j=1}^{m-1}\Delta\theta_j^2}{m(m-1)}} \tag{34}$$

*3.4. Algorithm Error of the Calibration Method of the Angular Positioning Deviation of the Rotary Table Based on the Laser Tracer Multi-Station Measurement System*

In the calibration process of the angular positioning deviation of the rotary table, it is assumed that the rotating axis of the rotary table is perpendicular to the plane of the table of the CMM. However, due to the squareness error of the X-axis or the Y-axis of the CMM, the rotating axis of the rotary table is not perpendicular to the plane of the CMM working plane. By comparing the working plane of the rotary table obtained by the coordinates fitting of the laser tracer multi-station with the CMM working plane, the algorithm error of the calibration method of the angular positioning deviation of the rotary table based on the laser tracer multi-station measuring system can be calculated.

The CMM working plane is $\beta$, as shown in Figure 4. The normal vector of the theoretical plane equation of the $\beta$ plane is $\overrightarrow{n_1} = (0,0,1)$. The normal vector of the $\gamma$ plane is $\overrightarrow{n_2} = (a,b,c)$. The angle between the $\beta$ plane and the $\gamma$ plane is $\alpha$, Then, we can get

$$\cos\alpha = \frac{c}{\sqrt{a^2 + b^2 + c^2}} \tag{35}$$

where $a$ is the coefficient of the X-axis direction of the fitting plane equation $\gamma$; $b$ is the coefficient of the Y-axis direction of the fitting plane equation $\gamma$; $c$ is the coefficient of the Z-axis direction of the fitting plane equation $\gamma$.

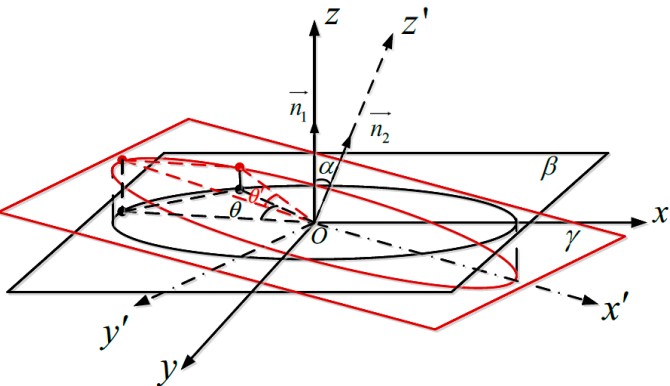

**Figure 4.** Schematic diagram of the angle between the rotating axis and the normal line of the working plane of the rotary table.

According to the geometric relationship, the algorithm error $\Delta$ of the calibration method of the angular positioning deviation of the rotary table based on the laser tracer multi-station measurement system can be expressed as

$$\Delta = \pm(\theta' - \theta) = \pm(ac\tan(\tan\theta\cos\alpha) - \theta) \tag{36}$$

## 4. Experimental Verification

### 4.1. Experimental System

In order to verify the calibration method of the angular positioning deviation of the rotary table based on the laser tracer multi-station measurement system proposed in this paper, a multistation measurement system of laser tracer was built (Figure 5). In the experiments, the measurement system was composed of the CMM (Leitz Infinity of Hexagon Company, Stockholm, Sweden), the high-precision rotary table (RT400, the angular positioning deviation = ±0.5″), and the laser tracer (Etalon Company, 07 2009 TR IF). The space range of the measured points in CMM was planned to be $450 \times 600 \times 450$ mm$^3$. In this space range, the number of planned measured points was $n = 64$, and the number of laser tracer stations was $m = 11$. The path planning of sampling points in the space range of the measured points is shown in Figure 6. There were four layers of paths. The height of each layer varied by 150 mm with the same moving path.

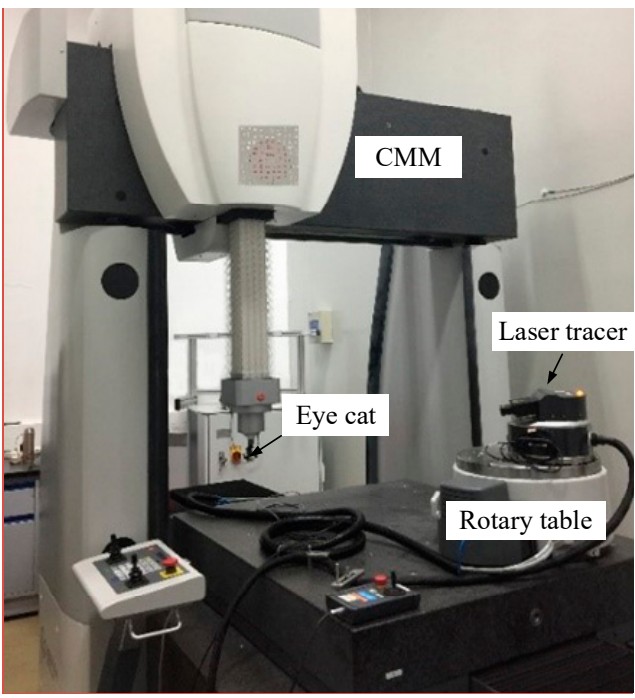

**Figure 5.** Experimental platform.

In the calibration process of the angular positioning deviation of the rotary table, considering the light-tracing problem of the laser tracer at some positions, the laser tracer was fixed on the position of the rotary table of −145°. This position was recorded as the first position $P_1$ of the laser tracer. The probe of CMM moved to $A_1$. The rotary table rotated counterclockwise by 30°. The laser tracer rotated together with the rotary table. During the rotation process of the laser tracer from the first position $P_1$ to the eleventh position $P_{11}$, the interferometric length $l_{1j}$ of the laser tracer was respectively recorded. Then, the probe of CMM was controlled to move from $A_1$ to $A_2$ along the planned path in the measurement space. During the rotation process of the laser tracer from the first position $P_1$ to the eleventh position $P_{11}$, the interferometric length $l_{2j}$ of the laser tracer was respectively recorded. Then, the probe of CMM was controlled to move from point $A_1$ to $A_i$ along the planned path in the measurement space. During the rotation process of the laser tracer from the first position $P_1$ to the eleventh position $P_{11}$, the interferometric length $l_{ij}$ was respectively recorded in turn. A total of 704 laser tracer interferometric lengths were recorded during the experiment.

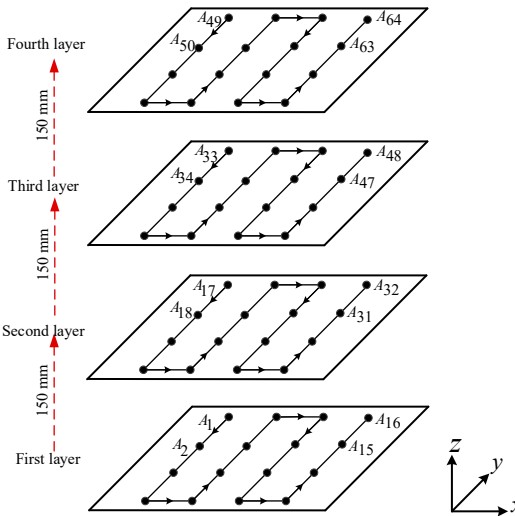

**Figure 6.** Path planning of sampling points in the measurement space.

## 4.2. Analysis of Experimental Results

### 4.2.1. Self-Calibration Coordinates of Laser Tracer Stations

The positioning precision of laser tracer is below 1 μm [18]. The precision requirement of the laser tracer multi-station measurement system [19] built in this paper was below 1 μm, $\varepsilon = 10^{-4}$. Through a series of experiments for the parameters, the coordinates of calibrated laser tracer stations were observed, and the optimal setting of the parameters was obtained as: $\beta = 0.55$ and $\sigma = 0.4$. In the measurement system, the initial value $\mu_i$ was set as $\|f_i(X_0, Y_0, Z_0, d_0)\|$, where $X_0 = -64.9938$, $Y_0 = -6.6256$, $Z_0 = 122.5133$, and $d_0 = 1128.5363$. Then $\mu_0 = 0.75$ was calculated. According to Equation (17), these parameters follow the rules of $\mu_i$ in L-M algorithm. According to the initial coordinate values $(X_0, Y_0, Z_0)$ of the laser tracer, the initial distance $d_0$ from laser tracer to the initial measured point, and the selected parameters in L-M algorithm, the coordinates $P_j(X_j, Y_j, Z_j)$ of the laser tracer and the distance $d_j$ from the laser tracer to the initial measured point were obtained by iteration (Table 1).

**Table 1.** Coordinates of laser tracer stations and their distance to the initial measured point (unit: mm).

| Coordinates | $X_j$ | $Y_j$ | $Z_j$ | $d_j$ |
| --- | --- | --- | --- | --- |
| 1 | −64.9938 | −6.6256 | 122.5146 | 1128.5363 |
| 2 | −53.0600 | −38.1824 | 122.5137 | 1128.5372 |
| 3 | −26.9493 | −59.5448 | 122.5127 | 1128.5363 |
| 4 | 6.3444 | −64.9883 | 122.5125 | 1128.5375 |
| 5 | 37.8969 | −53.0580 | 122.5123 | 1128.5373 |
| 6 | 59.2589 | −26.9510 | 122.5112 | 1128.5378 |
| 7 | 64.7067 | 6.3400 | 122.5104 | 1128.5377 |
| 8 | 52.7784 | 37.8971 | 122.5107 | 1128.5373 |
| 9 | 26.6701 | 59.2624 | 122.5109 | 1128.5382 |
| 10 | −6.6235 | 64.7100 | 122.5107 | 1128.5383 |
| 11 | −38.1803 | 52.7804 | 122.5108 | 1128.5379 |

### 4.2.2. Fitting Plane of Station Coordinates and Coordinates of Optimized Stations

The plane vector corresponding to the minimum singular value in the SVD transform is the normal vector of the fitting plane equation. The coefficients of the plane equation ($aX + bY + cZ + e = 0$) are $a = 1.9405 \times 10^{-5}$, $b = 1.5739 \times 10^{-5}$, $c = 0.9999$, and $e = -122.5119$, respectively. According to the results in Section 3.2, the SVD transform of the covariance matrix was used for plane fitting, and the fitting plane was obtained (Figure 7).

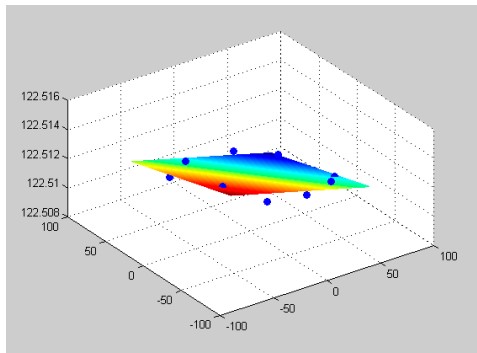

**Figure 7.** Schematic diagram of the fitting plane.

According to Equation (21) and the obtained values of the coefficients of the plane equation (*a*, *b*, *c*, *e*), $\lambda = -0.0013$ was calculated. The optimized coordinates of laser tracer stations are shown in Table 2.

**Table 2.** Optimized coordinates of laser tracer stations (unit: mm).

| Coordinates | $X_j$ | $Y_j$ | $Z_j$ |
|:---:|:---:|:---:|:---:|
| 1 | −64.9938 | −6.6256 | 122.5133 |
| 2 | −53.0600 | −38.1824 | 122.5136 |
| 3 | −26.9493 | −59.5448 | 122.5134 |
| 4 | 6.3444 | −64.9883 | 122.5128 |
| 5 | 37.8969 | −53.0580 | 122.5120 |
| 6 | 59.2589 | −26.9510 | 122.5112 |
| 7 | 64.7067 | 6.3400 | 122.5106 |
| 8 | 52.7784 | 37.8971 | 122.5103 |
| 9 | 26.6701 | 59.2624 | 122.5105 |
| 10 | −6.6235 | 64.7100 | 122.5110 |
| 11 | −38.1803 | 52.7804 | 122.5118 |

According to Equations (28)–(31), the coordinate of the center of the fitting circle $(x_c, y_c, z_c)$ is obtained as (−0.1434, −0.1396, 122.5119).

### 4.2.3. Angular Positioning Deviation of the Rotary Table

According to the calibration method of the angular positioning deviation of the rotary table based on the laser tracer multi-station measurement system proposed in this paper, the angular positioning deviations $\Delta\theta_j$ of the rotary table were obtained according to Equations (32) and (33) and provided in Table 3. The angular positioning deviation of the rotary table at each indexing angle is shown in Figure 8.

**Table 3.** Angular positioning deviation of the rotary table obtained by experiments.

| Stations | P2 | P3 | P4 | P5 | P6 | P7 | P8 | P9 | P10 | P11 |
|:---:|:---:|:---:|:---:|:---:|:---:|:---:|:---:|:---:|:---:|:---:|
| Angular positioning deviation $\Delta\theta_j$ /(unit: ″) | 5.9 | 0.7 | −0.4 | −9.9 | −9.2 | −7.2 | 1.0 | −0.1 | 1.0 | 2.9 |

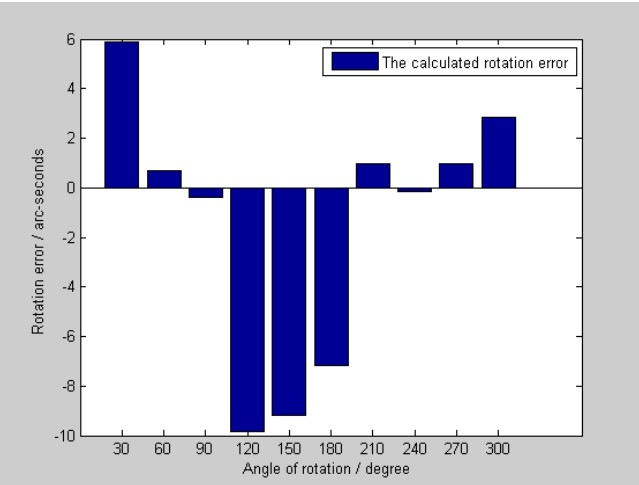

**Figure 8.** Angular positioning deviation of the rotary table at each indexing angle.

According to Equation (34), the angular positioning deviation of the rotary table based on laser tracer multi-station measurement is calibrated as

$$\bar{\sigma}_{\Delta\theta} = \pm\frac{1}{2}\sqrt{\frac{\sum\limits_{j=1}^{m-1}\Delta\theta_j^2}{m(m-1)}} = \pm0.9'' \tag{37}$$

### 4.2.4. Algorithm Error of the Calibration Method of The Angular Positioning Deviation of the Rotary Table Based on the Laser Tracer Multi-Station Measurement System

According to Equation (36), with the above experimental data, the algorithm error of the laser tracer multi-station measurement system for calibrating the positioning accuracy of the rotary table was calculated as

$$\Delta = \pm0.4'' \tag{38}$$

## 5. Conclusions

In this study, the laser tracer multi-station measurement system was used to calibrate the angular positioning deviation of the rotary table. The algorithm error of the high-precision rotary table method based on the laser tracer multi-station measurement system was mainly analyzed. A laser tracer multi-station measurement system was built. The rotary table was mounted on the working plane of CMM, and the laser tracer was placed on the rotary table. The rotary table rotated with the same angle intervals. With the redundancy equation of the laser tracer multi-station measurement system, the nonlinear least squares method was used to self-calibrate the coordinates of laser tracer stations. The coordinates of laser tracer stations were iterated with the Levenberg–Marquardt algorithm. The singular value decomposition (Singular Value Decomposition) of the covariance matrix was used to fit the m coordinates of laser tracer stations obtained from the calibration algorithm to obtain a plane. Then *m* coordinates of laser tracer stations were projected onto the fitting plane to obtain the optimized coordinates of laser tracer stations. The circle was fitted with the coordinates of optimized laser tracer stations, and then the coordinates of the center of the fitting circle were solved. Then, the geometric relationship between the coordinates of optimized laser tracer stations and the center of the obtained fitting circle was established, and the actual rotation angle of the rotary table was obtained. In this way, the calibration of the angular positioning deviation of the rotary table was realized.

The experimental parameters were as follows. The spatial range of measured points given by CMM was $450 \times 600 \times 450 \text{ mm}^3$; the number of measured points was 64; the number of laser tracer

stations was 11; the angle range of rotary table was [−145°,155°]; the rotation angle interval of the rotary table was 30°. Under the above experimental conditions, the conclusions were drawn as follows: The angular positioning deviation of the rotary table was as low as ±0.9″ compared with other work. Besides, the simple structure made it possible to meet the requirements of fast and efficient calibration. The algorithm error of the laser tracer multi-station measurement system for calibrating angular positioning deviation of the rotary table was ±0.4″. The experiment proved the feasibility of the proposed calibration method. In actual industrial measurement, the angular positioning deviation demand of high-precision rotary tables is low.

Although there is a limitation for this case that the rotary table should not be linked with the CMM, it is especially suitable for calibration of angular positioning deviation of large high-precision rotary tables. It also makes a theoretical preparation for the calibration of four-axis machine tools.

**Author Contributions:** Data curation, H.C. and B.J.; Investigation, H.C.; B.J. and H.L.; Methodology, H.C. and B.J.; Resources, Z.S.; Writing—original draft, S.Z., H.S. and Y.S.; Writing—review & editing, S.Z., H.S. and Y.S.

**Funding:** The research was funded by National Key Research and Development Program "Making basic technology and key components" special project under Grant No. 2018YFB2001402, Natural Science Foundation of Beijing, China under Grant No.3182005, National Natural Science Foundation of China under Grant No. 51635001.

**Conflicts of Interest:** The authors declare no conflict of interest.

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
