# Peer review of "Calibration Method for Angular Positioning Deviation of a High-Precision Rotary Table Based on the Laser Tracer Multi-Station Measurement System"

_applsci, doi:10.3390/app9163417_

Round 1

Reviewer 1 Report

The paper topic is interesting and paper can be published but only after the above suggestions:

Abstract: The abstract is expected to include a brief digest of the research, that is, new methods, results, concepts, and conclusions only. The abstract needs to be more focused and achievements needs mentioned clearly. At the moment abstract is more like an introduction than abstract. Please add some information from the conclusion (quantifications).

Introduction based on old references from one side of world. This part of paper is a weak point. I personally feel that this part of paper is not concise enough from a reader’s perspective. Introduction must provide a comprehensive critical review of recent developments in a specific area or theme that is within the scope of the journal (metrology), not only a list of published studies or a bibliometric one. Introduction is expected to have an extensive literature review followed by an in-depth and critical analysis of the state of the art. Authors do not write introduction in the context of Instrumentation and Measurement. Authors should present their work properly within the existing metrology literature; i.e., papers published in the metrology and measurement journals, and compare their work with these latest related work. This comparison should be done analytically (in the introduction or related work section), experimentally (in the performance evaluation section), or both. References section should be extensive about information connecting with metrology equipment and measurement methods. I suggest add information to better describe what other researchers have done in this area. I suggest add important and new articles from this field.

Fig. 5 do not contribute anything important to the paper and should be removed.

In table 1 authors wrote dimension 122.5146 mm. The measurement device (typical) has a lower interaction. So how reliable are the data presented? Not to four decimal places!

The discussion is shallow and needs more details, the observations and future trends. This chapter should be connected with others published papers.

Some of the bullet points on the conclusion are simplistic;  Please try to emphasize your novelty, put some quantifications, and comment on the limitations. This is a very common way to write conclusions for a learned academic journal. The conclusions should highlight the novelty and advance in understanding presented in the work.

Reviewer 2 Report

The paper concerns the calibration of the rotary table of high-precision. The issue is important from the industrial measurement point of view. Highlighting it can help several rotary table users in their work. Though I have some comments regarding how the task has been presented.

The introduction includes information about existing systems though the authors do not provide their accuracy in numbers, which would be very helpful.

The concept a s a whole have been described but it is not clear why it has been arranged and implemented exactly in the presented way. In addition the system has not been clearly illustrated by the included figures. Figures 1 and 2 do not sufficiently reflect the description (lines 79-89). Besides they are too small and not very informative. They do not include details. As a result following the main concept of the approach requires some efforts.

The sequence of mathematical operations appeared to be correct. The usage if the Levenberg-Marquardt algorithm is justified. However, the authors say that the accuracy of the laser tracer is at the level of 1 µm-below the system requirements. Since the data processing includes two types of fittings i.e. a plane and circle it is possible that the accuracy will deteriorate significantly. In this context a more extended explanation would be expected.

Figures 4 and 5 are too small and Figure 6 should be better explained.

The conclusion again lacks a comparison with other approaches with numbers.

I think that the paper is interesting and certainly worth publishing but the mentioned above improvements are necessary.
